# Replication Study: *Fusobacterium nucleatum* infection is prevalent in human colorectal carcinoma

**John Repass, Reproducibility Project: Cancer Biology***

ARQ Genetics, Bastrop, United States

**Abstract** As part of the Reproducibility Project: Cancer Biology, we published a Registered Report (Repass et al., 2016), that described how we intended to replicate an experiment from the paper 'Fusobacterium nucleatum infection is prevalent in human colorectal carcinoma' (Castellarin et al., 2012). Here we report the results. When measuring *Fusobacterium nucleatum* DNA by qPCR in colorectal carcinoma (CRC), adjacent normal tissue, and separate matched control tissue, we did not detect a signal for *F. nucleatum* in most samples: 25% of CRCs, 15% of adjacent normal, and 0% of matched control tissue were positive based on quantitative PCR (qPCR) and confirmed by sequencing of the qPCR products. When only samples with detectable *F. nucleatum* in CRC and adjacent normal tissue were compared, the difference was not statistically significant, while the original study reported a statistically significant increase in *F. nucleatum* expression in CRC compared to adjacent normal tissue (Figure 2; Castellarin et al., 2012). Finally, we report a meta-analysis of the result, which suggests *F. nucleatum* expression is increased in CRC, but is confounded by the inability to detect *F. nucleatum* in most samples. The difference in *F. nucleatum* expression between CRC and adjacent normal tissues was thus smaller than the original study, and not detected in most samples.

DOI: https://doi.org/10.7554/eLife.25801.001

**\*For correspondence:**
tim@cos.io;
nicole@scienceexchange.com

**Group author details:**
Reproducibility Project: Cancer Biology See page 9

## Introduction

The Reproducibility Project: Cancer Biology (RP:CB) is a collaboration between the Center for Open Science and Science Exchange that seeks to address concerns about reproducibility in scientific research by conducting replications of selected experiments from a number of high-profile papers in the field of cancer biology (*Errington et al., 2014*). For each of these papers a Registered Report detailing the proposed experimental designs and protocols for the replications was peer reviewed and published prior to data collection. The present paper is a Replication Study that reports the results of the replication experiments detailed in the Registered Report (*Repass et al., 2016*), for a paper by Castellarin et al., and uses a number of approaches to compare the outcomes of the original experiments and the replications.

In 2012, Castellarin et al. reported that overall abundance of *Fusobacterium nucleatum* (*F. nucleatum*) RNA was increased by approximately 79 fold in colorectal carcinoma (CRC) as compared to adjacent normal biopsies, as determined by RNA sequencing. Next, they measured *F. nucleatum* DNA abundance in 99 CRC and adjacent normal tissue biopsies from a patient cohort and found that the presence of *F. nucleatum* DNA was 415 times greater in CRC tissue than adjacent normal tissue. These results, combined with earlier studies, provided evidence for a link between tissue-associated bacteria and tumorigenesis.

The Registered Report for the paper by *Castellarin et al. (2012)* described the experiments to be replicated (Figure 2), and summarized the current evidence for these findings. Since that publication there have been additional studies investigating *F. nucleatum* abundance in CRC. *F. nucleatum*

DNA abundance was reported to be enriched in 88 out of 101 CRC tissue samples compared to adjacent normal tissues in Chinese patients (*Li et al., 2016*). The proportion of CRC tissue that were found to be *F. nucleatum* DNA positive increased from rectal cancers to cecal cancers (*Gao et al., 2015*; *Mima et al., 2016a*), increased according to histological grade (*Ito et al., 2015*), and varied depending on diet (*Mehta et al., 2017*). An association of *F. nucleatum* DNA with clinical outcome in CRC patients has also been investigated, but with mixed outcomes. In one study, *F. nucleatum* positive cases resulted in a worse outcome (*Mima et al., 2016b*), while two separate studies found no association with cancer-specific mortality (*Ito et al., 2015*; *Mima et al., 2015*).

Many of these recent studies utilized the same TaqMan primer/probe set as *Castellarin et al. (2012)* and reported 12–13% of CRC samples as positive for *F. nucleatum* (*Mehta et al., 2017*; *Mima et al., 2016a*; *2016b*; *Mima et al., 2015*), compared to 3.4% positive in adjacent normal tissue (*Mima et al., 2015*). Another study, which utilized a custom-made TaqMan assay, reported 56% of CRC samples as positive, and that a subset of these positive CRC samples had lower *F. nucleatum* expression in the adjacent normal tissue (*Ito et al., 2015*). Ito and colleagues also examined *F. nucleatum* expression in normal mucosa samples and reported 15% were positive, but with low expression of *F. nucleatum* (*Ito et al., 2015*). Importantly, these studies utilized formalin-fixed paraffin-embedded (FFPE) tissue instead of fresh frozen tissue samples. While FFPE has the advantage of preserving morphology, it makes analysis of biomolecules, particularly nucleic acids, challenging due to formation of crosslinks as compared to fresh frozen tissue (*Bradley et al., 2015*; *Van Allen et al., 2014*).

In addition to using quantitative PCR (qPCR), *F. nucleatum* has also been visualized within CRC tissue using fluorescence in situ hybridization (FISH) (*Kostic et al., 2013*; *2012*; *Li et al., 2016*; *McCoy et al., 2013*). Another method of detecting bacterial infection is through a humoral immune response. *F. nucleatum* specific antibodies were detectable in samples from CRC patients and have been reported as a potential diagnostic biomarker for CRC (*Wang et al., 2016*). Finally, using pyro-sequencing, Gao and colleagues reported a higher abundance of bacteria from the genus *Fusobacterium* between CRC patients and separate healthy individuals (10.08% vs 0.01%, respectively) (*Gao et al., 2015*).

The outcome measures reported in this Replication Study will be aggregated with those from the other Replication Studies to create a dataset that will be examined to provide evidence about reproducibility of cancer biology research, and to identify factors that influence reproducibility more generally.

## Results and discussion

### Quantitative PCR of *F. nucleatum* DNA abundance from colorectal carcinoma, adjacent normal tissue, and matched normal human colon tissue

We sought to independently replicate an experiment testing the hypothesis that *F. nucleatum* is overrepresented in CRC tissue compared to adjacent normal tissue by qPCR. This experiment is similar to what was reported in Figure 2 of *Castellarin et al. (2012)*. While it is common practice to use adjacent normal tissue as a control to reduce the effect of genetic background, it is widely accepted that adjacent normal tissue, while observationally normal, may have genetic alterations that make it distinct from very distant somatic tissue (*Braakhuis et al., 2004*). To this end, gene expression profiling in adjacent normal tissue has even been used to predict recurrence in patients with rectal cancer (*Schneider et al., 2006*) and to predict recurrence as well as overall survival time in breast cancer patients (*Troester et al., 2016*). Thus, as an additional control, this replication attempt included a group of normal colorectal tissues from age/ethnicity matched patients. Extracted genomic DNA was analyzed for abundance of *F. nucleatum*, which was determined using the same primers as the original study targeting an *F. nucleatum* specific gene. The human *PGT* gene (updated gene symbol: solute carrier organic anion transporter family member 2A1 (*SLCO2A1*)) served as the reference control, with the ratio of these two genes giving an estimate of the ratio of *F. nucleatum* DNA to human DNA in each sample. Two independent qPCR runs were performed.

The cycle threshold (Ct) values, a measure of the concentration of the target sequence in the PCR reaction, observed in this replication attempt ranged from 22.4 to 40 for adjacent normal tissue, 22.1 to 40 for CRC tissue, and 21.8 to 40 for matched normal tissue (*Figure 1—figure supplement 1*, *Figure 1—figure supplement 2*). This compares to the original study that reported Ct values that ranged from 25.5 to 40 for adjacent normal tissue and 21.4 to 40 for CRC tissue. However, these ranges include all Ct values observed (*PGT* and *F. nucleatum*). We observed that while the raw Ct values for *PGT* were well within the normal acceptable range (<30) (*Karlen et al., 2007*), the raw Ct values for *F. nucleatum* were very high across both independent qPCR runs for all samples. For *PGT*, the first independent run (*Figure 1—figure supplement 1A*) yielded a median raw Ct value of 22.7 (IQR = 22.6–22.9) in adjacent normal tissue, 22.8 (IQR = 22.6–22.8) in CRC tissue, and 22.8 (IQR = 22.7–23.0) in matched normal tissue. However, *F. nucleatum* did not give a signal in most samples. In these samples, the Ct was set to 40, even though no amplicon was observed (also known as 'non-detects'). While others have suggested methods to reduce the bias introduced when non-detects are set to Ct values of 40, such as setting the value to 35 (*McCall et al., 2014*), or using imputation methods and hierarchical models to deal with non-detects (*McCall et al., 2014*; *Boyer et al., 2013*), the approach should be based on evidence (*Caraguel et al., 2011*) and introduced prior to data collection, such as in a pre-registered analysis plan, to minimize confirmation bias (*Wagenmakers et al., 2012*). Furthermore, in the samples in which *F. nucleatum* was detectable, it was often at the edge of detectability, requiring more than 30 cycles of PCR to be detected (*Figure 1—figure supplement 1B*). This makes the data difficult to interpret considering the noise associated with very high Ct values. A similar observation was made with data from the second independent run (*Figure 1—figure supplement 2B*). There was high concordance for the *PGT* Ct values between the two runs with a correlation coefficient (ρ) of 0.84, 0.93, and 0.96 for adjacent normal, CRC, and matched normal tissue, respectively. Similar high concordance was observed for the *F. nucleatum* Ct values with a ρ of 0.99, 0.96, and 0.81 for adjacent normal, CRC, and matched normal tissue, respectively.

Although not pre-specified, to assess if the signal for *F. nucleatum* from the qPCR assay produced a specific product for *F. nucleatum*, we sequenced the amplicons generated. When we examined the 10 adjacent normal and 16 CRC tissue samples that gave a signal for *F. nucleatum* from the qPCR assay, we observed that at higher Ct values (Ct >35) a specific sequence was detected in some samples (3 CRC), while others only produced non-specific amplification (4 adjacent normal; 6 CRC). The non-specific amplification observed could be due to poor DNA quality or poor cleanup of the DNA samples during the sequencing process. In all of the samples with Ct values less than 35, specific sequences were observed (6 adjacent normal; 7 CRC). Interestingly, there were six paired samples, all with Ct values less than 35, where a specific sequence was observed in the adjacent normal as well as in the CRC tissue. These samples were further analyzed to determine if the relative abundance of *F. nucleatum* DNA (normalized to *PGT*) between the tissues were different (*Figure 1*). Performing an exploratory analysis on the mean normalized expression of *F. nucleatum* DNA (normalized to *PGT*) from the two independent runs, we found that the median fold change in *F. nucleatum* DNA between CRC and adjacent normal tissue [n = 6] was 1.42 (IQR = 0.72–2.80) with the mean fold change equal to 2.78 [SD = 3.46]. This difference was not statistically significant (two-tailed paired *t* test: $t(5) = 1.067$, $p = 0.335$, $r = 0.43$, 95% CI [−0.59, 0.92]) when only samples in which *F. nucleatum* was detected in both CRC and adjacent normal tissue was considered. Considering only the data in which a specific product from the qPCR assay was able to be confirmed, *F. nucleatum* expression was higher in the CRC tissue compared to the adjacent normal tissue in 8 of the 40 examined samples, while expression was higher in the adjacent normal tissue compared to CRC tissue in two samples.

To facilitate a direct comparison of these results to the original study we included all of the samples in the analysis, as was done in the original study, and specified in the confirmatory analysis plan of the Registered Report (*Repass et al., 2016*). Samples in this case were identified positive based upon whether a product was amplified in the qPCR reaction, irrespective of whether the amplicon was confirmed to be *F. nucleatum* by sequencing. While allowing us to compare qPCR signal intensity among samples, this approach also introduces error into the comparison because, as noted above, we know that some of the amplification products from the qPCR assay were non-specific. The relative abundance of *F. nucleatum* DNA (normalized to *PGT*) between CRC and adjacent normal tissue was determined for each independent run (*Figure 2—figure supplement 1*) and

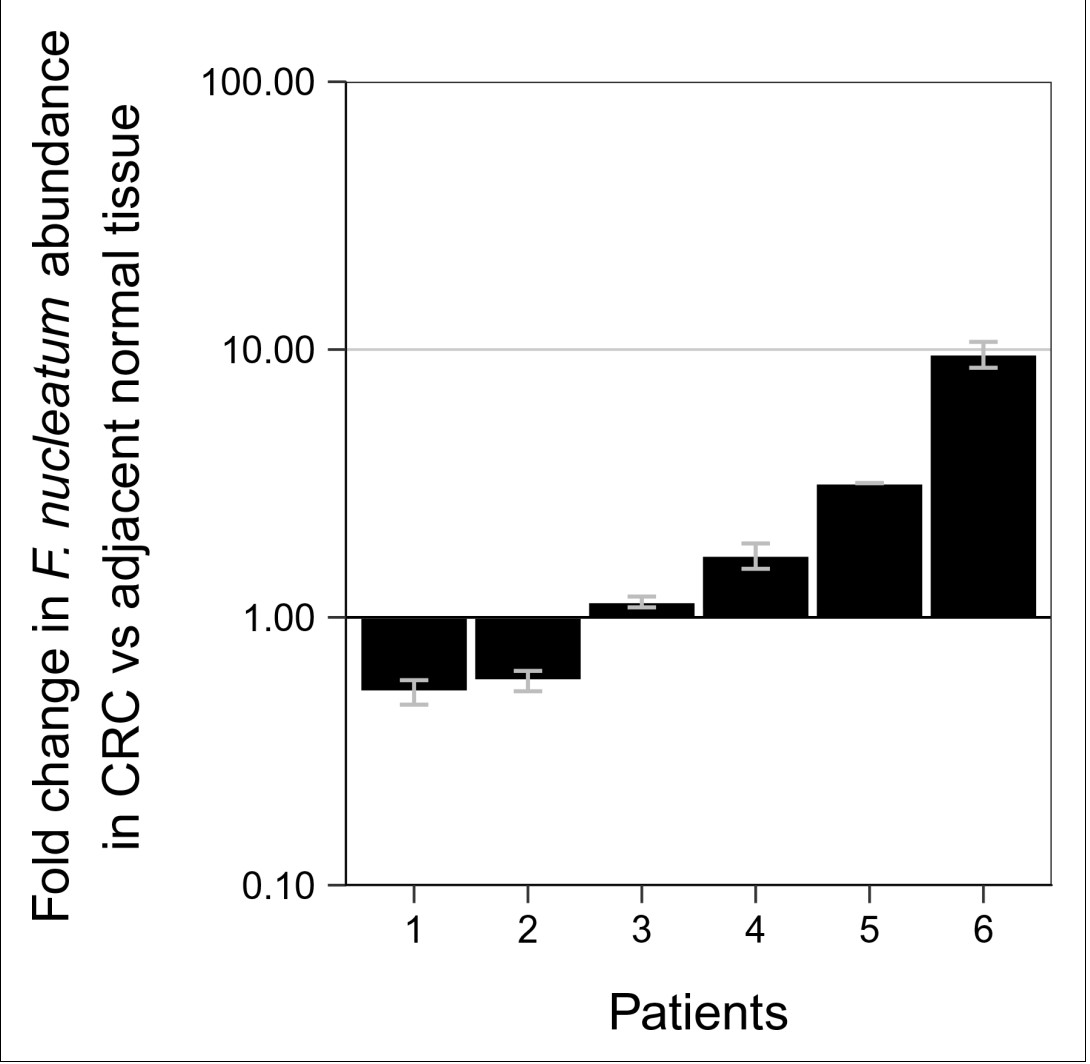

**Figure 1.** Relative abundance of *F. nucleatum* in colorectal carcinoma versus adjacent normal biopsies in samples with detectable *F. nucleatum*. Only samples in which *F. nucleatum* was detected in both colorectal carcinoma (CRC) and adjacent normal tissue by quantitative PCR (qPCR) and confirmed by sequencing of the qPCR product are shown (n = 6). The mean relative abundance of *F. nucleatum* (normalized to *PGT* expression) in CRC tissue versus adjacent normal tissue of both independent runs is reported for each patient sample and error bars represent *SEM*. The y-axis represents mean fold gene expression change ($2^{-\Delta\Delta Ct}$) while the x-axis represents patient samples. Exploratory analysis: two-tailed paired *t* test: *t*(5) = 1.067, *p* = 0.335, *r* = 0.43, 95% CI [−0.59, 0.92]. Additional details for this experiment can be found at https://osf.io/rb4yq/.
DOI: https://doi.org/10.7554/eLife.25801.002
The following figure supplements are available for figure 1:

**Figure supplement 1.** First independent qPCR run.
DOI: https://doi.org/10.7554/eLife.25801.003
**Figure supplement 2.** Second independent qPCR run.
DOI: https://doi.org/10.7554/eLife.25801.004

averaged for the analysis (*Figure 2*). We found that the median fold change in *F. nucleatum* DNA between CRC and adjacent normal tissue [n = 40] was 1.13 (IQR = 0.91–2.84) with the mean fold change equal to 4.97 [*SD* = 14.2]. This compares to the original study, which reported an estimated median fold change of *F. nucleatum* DNA in CRC to adjacent normal tissue [n = 99] of 3.15 (IQR = 1.04–14.49) with the mean fold change equal to 378.9 [*SD* = 1980] (*Castellarin et al., 2012*). Analysis of the mean normalized expression of *F. nucleatum* DNA (normalized to *PGT*) from the two

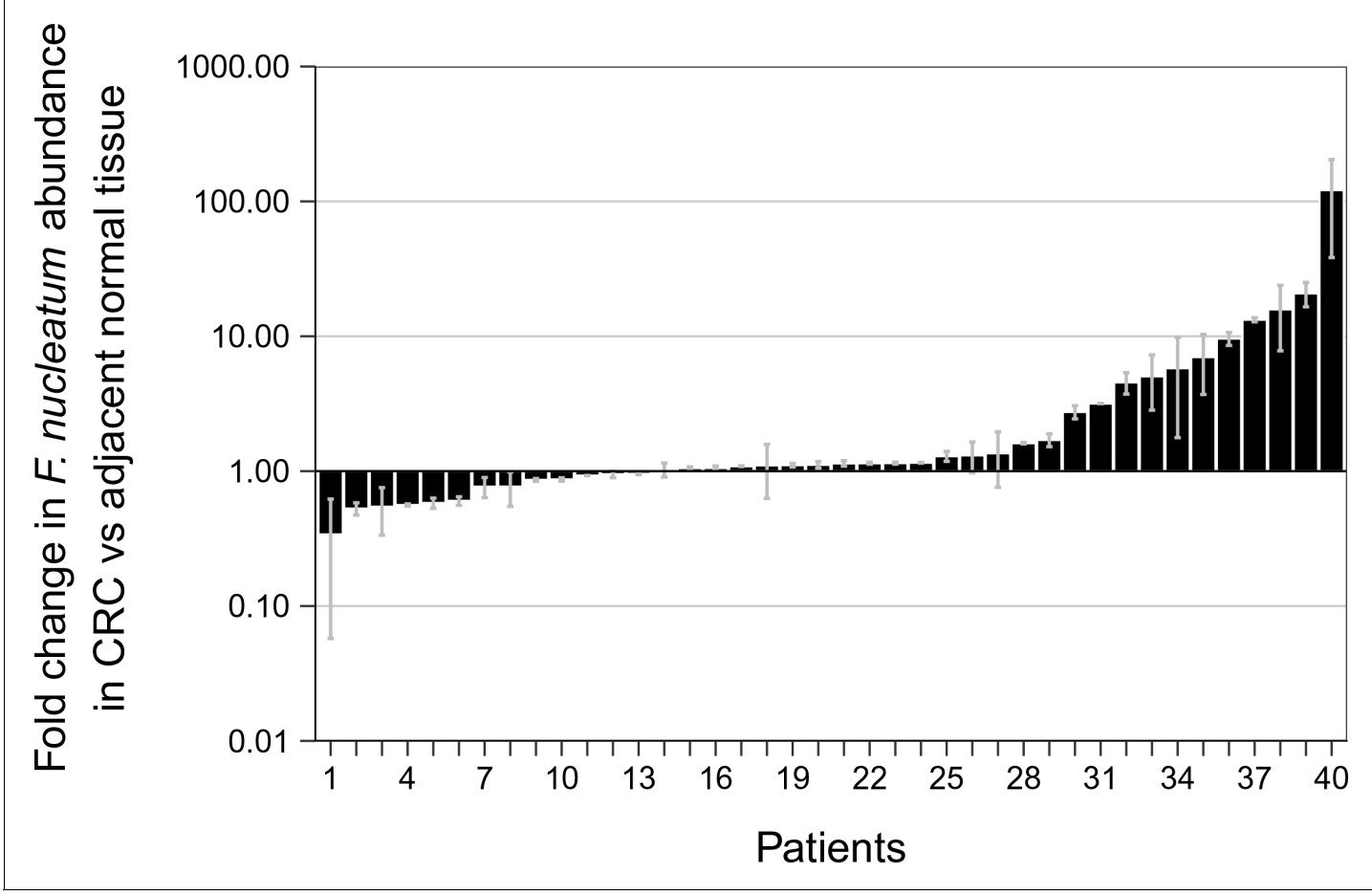

**Figure 2.** Relative abundance of *F. nucleatum* by qPCR in colorectal carcinoma versus adjacent normal biopsies. Fold change values are shown for all paired specimens based on differences in Ct values, irrespective of whether the qPCR products were confirmed to be specific or non-specific upon sequencing. Tissue was collected from colorectal carcinoma (CRC) and adjacent normal tissue (n = 40). qPCR was performed independently two times and averaged. The mean relative abundance of *F. nucleatum* (normalized to *PGT* expression) in CRC tissue versus adjacent normal tissue of both independent runs is reported for each patient sample and error bars represent SEM. The y-axis represents mean fold gene expression change ($2^{-\Delta\Delta Ct}$) while the x-axis represents patient samples. Two-tailed Wilcoxon signed-rank test: *Z* = 2.14, *p* = 0.032. Additional details for this experiment can be found at https://osf.io/rb4yq/.

DOI: https://doi.org/10.7554/eLife.25801.005

The following figure supplement is available for figure 2:

**Figure supplement 1.** Independent qPCR runs.

DOI: https://doi.org/10.7554/eLife.25801.006

independent runs was statistically significant (two-tailed Wilcoxon signed-rank test: *Z* = 2.14, *p* = 0.032), which suggests that *F. nucleatum* DNA is over represented in CRC compared to adjacent normal tissue. Collectively, 5% (2 out of 40 samples) of the matched normal tissue samples, 25% (10 out of 40 samples) of the adjacent normal tissue, and 40% (16 out of 40 samples) of the CRC tissue gave qPCR products for *F. nucleatum* (i.e. amplification of a product after fewer than 40 cycles in both independent repeats, some of which were confirmed to have amplified *F. nucleatum* sequence and some that were non-specific, as noted above). There were three samples that gave qPCR products for *F. nucleatum* in the adjacent normal tissue and not in the CRC tissue, nine samples that gave qPCR products in the CRC tissue and not in the adjacent normal tissue, and seven samples that gave qPCR products in the adjacent normal as well as in the CRC tissue. Considering all of the data in which a product was amplified in the qPCR reaction, and irrespective of whether the qPCR products were confirmed to be specific or non-specific upon sequencing, *F. nucleatum* expression was higher

in the CRC tissue compared to adjacent normal tissue in 13 of the 40 examined samples, while expression was higher in the adjacent normal tissue compared to CRC tissue in six samples. The difference in the observed fold change between this replication attempt and the original study could be due to a number of factors between the studies. Particularly with so many samples around the threshold of detection for *F. nucleatum*, very small changes in signal can lead to large changes in fold-change values (*Cui et al., 2015*).

## Meta-analyses of original and replicated effects

We performed a meta-analysis using a random-effects model to combine each of the effects from the original study and this replication described above as pre-specified in the confirmatory analysis plan (*Repass et al., 2016*). To directly compare and combine the results of both studies, we used the qPCR results for this replication irrespective of whether the amplicon was confirmed to be *F. nucleatum* by sequencing. To provide a standardized measure of the effect, a common effect size was calculated for the original and replication studies. The effect size $r$ is a standardized measure of the correlation (strength and direction) of the association between two variables, in this case tissue type and normalized *F. nucleatum* DNA levels. Since *F. nucleatum* was not detected in most samples, this confounds the effect size for this meta-analysis and makes it difficult to compare the replication data to the original study, which did not report the number of samples below the limit of detection.

The comparison of CRC tissue to adjacent normal tissue resulted in $r = 0.45$, 95% CI [0.28, 0.60] for the effect size estimated *a priori* from the *p*-value and sample size reported in Figure 2 of the original study (*Castellarin et al., 2012*). This compares to $r = 0.24$, 95% CI [−0.08, 0.51] for the comparison of all of the CRC to adjacent normal tissue samples reported in this study, which spans zero and implies that the null hypothesis cannot be rejected. A random effects meta-analysis (*Figure 3*) of both the replication and original effects resulted in $r = 0.38$, 95% CI [0.17, 0.56], which was statistically significant ($p = 5.86 \times 10^{-4}$). Using the estimate of the effect size of one study, as well as the associated uncertainty (i.e. confidence interval), and comparing it to the effect size of the other study provides another approach to compare the original and replication results (*Errington et al., 2014*; *Valentine et al., 2011*). Importantly, the width of the confidence interval for each study is a reflection of not only the confidence level (e.g. 95%), but also variability of the sample (e.g. *SD*) and sample size. Thus, both studies, observed higher *F. nucleatum* levels in CRC tissue. The point estimate of the replication effect size was not within the confidence interval of the original result; however, the point estimate of the original effect size was within the confidence interval of the replication.

This direct replication provides an opportunity to understand the present evidence of these effects. Any known differences, including reagents and protocol differences, were identified prior to conducting the experimental work and described in the Registered Report (*Repass et al., 2016*). However, this is limited to what was obtainable from the original paper, which means there might be particular features of the original experimental protocol that could be critical, but unidentified. So while some aspects, such as method of qPCR and primer and probe sequences were maintained, others were unknown or not easily controlled for, including similarities and differences in patient characteristics (*Klevorn and Teague, 2016*), as well as influence of diet on the gut microbiota (*Mehta et al., 2017*). Whether these or other factors influence the outcomes of this study is open to hypothesizing and further investigation, which is facilitated by direct replications and transparent reporting.

## Materials and methods

As described in the Registered Report (*Repass et al., 2016*), we attempted a replication of the experiment reported in Figure 2 of *Castellarin et al. (2012)*. A detailed description of all protocols can be found in the Registered Report (*Repass et al., 2016*). Additional detailed experimental notes, data, and analyses are available on OSF (RRID:SCR_003238) (https://osf.io/v4se2/; *Repass et al., 2017*). This includes the R Markdown file (https://osf.io/fmp6u/) that was used to compose this manuscript, which is a reproducible document linking the results in the article directly to the data and code that produced them (*Hartgerink, 2017*).

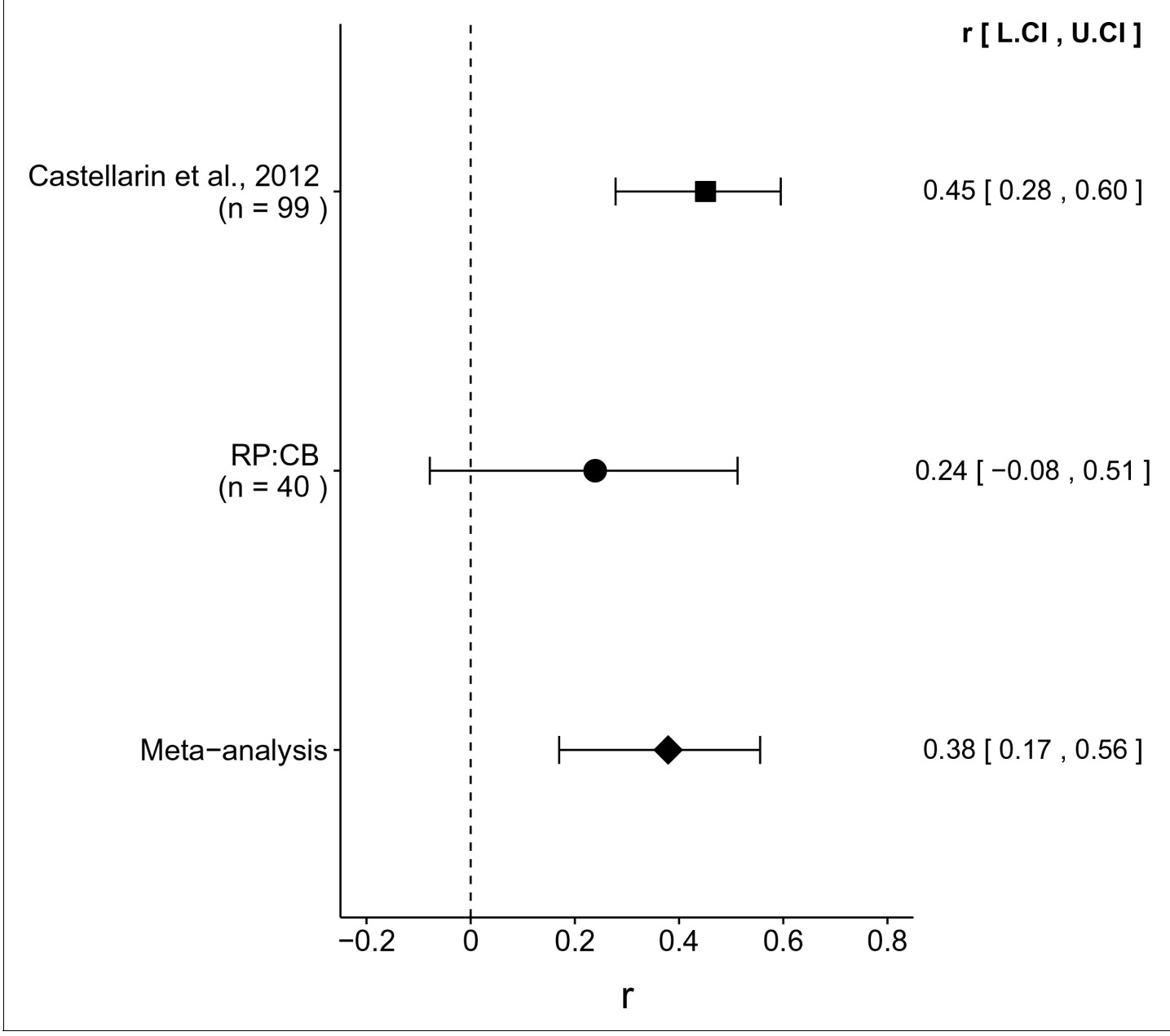

**Figure 3.** Meta-analyses of each effect. Effect size (r) and 95% confidence interval are presented for *Castellarin et al. (2012)*, this replication attempt (RP:CB), and a meta-analysis to combine the two effects. To directly compare and combine the results of both studies, the qPCR results for RP:CB were used irrespective of whether the amplicon was confirmed to be *F. nucleatum* by sequencing. The effect size *r* is a standardized measure of the correlation (strength and direction) of the association between tissue type and normalized *F. nucleatum* DNA levels, with a larger positive value indicating CRC tissue is correlated with a higher *F. nucleatum* expression level. Sample sizes used in *Castellarin et al. (2012)* and this replication attempt are reported under the study name. Random effects meta-analysis of the abundance of *F. nucleatum* in CRC tissue compared to adjacent normal tissue (meta-analysis $p = 5.86 \times 10^{-4}$). Additional details for this meta-analysis can be found at https://osf.io/kup8d/.
DOI: https://doi.org/10.7554/eLife.25801.007

## Clinical specimens

Patient tissue samples were obtained from iSpecimen (Lexington, Massachusetts) as flash-frozen in liquid nitrogen shortly after harvest. This included 40 CRC samples along with adjacent normal tissue from the same patient, as well as 40 age, sex, and ethnicity matched non-diseased control colorectal tissues. Patient phenotypes (age, gender, ethnicity, diagnosis, result, and histopathology report) are

available at https://osf.io/tc3jc/. The original study did not report patient phenotypes, thus it is unknown how the patient populations compared between the two studies. Approval was obtained from the iSpecimen institutional review board (protocol # 2011–332) and shared samples and data were de-identified for this study.

## Quantitative PCR

Genomic DNA was isolated from all samples using the Gentra Puregene genomic DNA extraction kit (Qiagen, cat# 158389) with Proteinase K (Roche, cat# 0311587900) as outlined in the Registered Report (full protocol details are available at https://osf.io/y8eum/). Each sample was approximately the same size, with the amount of Proteinase K and other additives used based on volume calculations and assumed tissue density of 1.04 g/cm$^3$. Following an initial qPCR run, genomic DNA was re-purified using the Qiagen DNeasy Blood and Tissue Kit (Qiagen, cat# 69504) according to the manufacturer's instructions and the amount of starting material was increased in an attempt to optimize the detection of *F. nucleatum*. DNA was quantified using a Nanodrop spectrophotometer (Thermo Fisher Scientific, cat# ND-1000). qPCR was performed on the ABI 7900HT Fast Real Time PCR System (Applied Biosystems, cat# 4351405) using assays specific for each gene of interest as outlined in the Registered Report (*Repass et al., 2016*). For both assays, each reaction well contained 10 µL of TaqMan Universal Master Mix II (Applied Biosystems, cat# 4440039), 60 ng (first independent run) or 100 ng (second independent run) total DNA, at 30 ng/µL, and 1 µL of each assay (final forward/reverse primer concentration 900 nM and probe concentration 250 nM) in a reaction volume of 20 µL. Cycling conditions were as follows: 50°C for 2 min for UNG activation, 95°C for 10 min for polymerase activation, followed by 40 cycles of 95°C for 15 s and 60°C for 1 min. Ct values were obtained using Sequence Detection System software from Applied Biosystems, version 2.4. The reaction for the *F. nucleatum* assay, using six serial logarithmic dilutions of purified genomic DNA from *F. nucleatum* subsp. *Nucleatum* Strain VPI 4355 (ATCC, cat# 25586D-5), was found to have an efficiency of 99% with a sensitivity to detect at least 1 pg DNA. The reaction for the *PGT* assay, using four serial logarithmic dilutions of human genomic DNA, was found to have an efficiency of 96% with a sensitivity to detect at least 100 pg DNA. Standard curve data are available at https://osf.io/32e8q/. For all qPCR runs, the *F. nucleatum* assay had a baseline of cycles 3–30 and a threshold of 0.2 and the *PGT* assay had a baseline of cycles 3–20 and a threshold of 0.1. For qPCR preprocessing all nondetects were set to 40. Technical duplicates were averaged for each sample. Fold difference in *F. nucleatum* abundance in tumor versus normal tissue was determined using the ΔΔCt method (*Pfaffl, 2001*). For each run, ΔCt (*F. nucleatum* expression normalized to *PGT* expression) was calculated for each tissue sample and then the fold difference ($2^{-\Delta\Delta Ct}$) in *F. nucleatum* abundance in tumor versus adjacent normal tissue was determined. There was high concordance of normalized expression between the two runs with a correlation coefficient (ρ) of 0.99, 0.98, 0.97 for adjacent normal, CRC, and matched normal tissue, respectively. The mean ΔCt was calculated for each sample across the two independent qPCR runs and used for statistical analysis. Additional information, including all raw qPCR data, is available at https://osf.io/rb4yq/.

## Sequencing PCR products

Multiple methods for sequencing PCR products from *F. nucleatum* amplicons were attempted, utilizing both patient tissue samples and a control of purified genomic DNA from *F. nucleatum* subsp. *Nucleatum* Strain VPI 4355. In the first approach, PCR products were amplified from the appropriate template using AmpliTaq Gold DNA Polymerase (Thermo Fisher Scientific, cat# 4311806) and *F. nucleatum* specific forward and reverse primers and amplification conditions as described above. No products were seen by agarose gel electrophoresis. In a second approach, TaqMan Universal Master Mix with UNG (Thermo Fisher Scientific, cat# 4440038) with the same primers and amplification conditions as above were attempted. PCR products of the correct size (~125 bp) were seen by agarose gel electrophoresis and were then used with the TOPO TA Cloning Kit (Thermo Fisher Scientific, cat# K457540) in an attempt to generate plasmids containing the *F. nucleatum*-derived PCR products for Sanger sequencing analysis. Several attempts resulted in no colonies positive for the correct insert. Finally, PCR products were generated using the same approach described in the second attempt (with TaqMan) and analyzed by direct Sanger sequencing using the *F. nucleatum* specific forward amplification primer. Automated DNA sequencing was performed using industry standard,

capillary-based Applied Biosystems 3730/3730XL DNA analyzers (Applied Biosystems, Inc., Foster City, California) and BigDYe Terminator 3.1 cycle sequencing kit (Thermo Fisher Scientific, cat# 4337455).

## Statistical analysis

Statistical analysis was performed with R software (RRID:SCR_001905), version 3.4.2 (*R Core Team, 2017*). All data, csv files, and analysis scripts are available on the OSF (https://osf.io/v4se2/). Confirmatory statistical analysis was pre-registered (https://osf.io/2wmfb/) before the experimental work began as outlined in the Registered Report (*Repass et al., 2016*). All additional analysis reported is exploratory. Data were checked to ensure assumptions of statistical tests were met. A meta-analysis of a common original and replication effect size, with confidence intervals determined by Fisher's z' transformation (*Rosenthal and DiMatteo, 2001*), was performed with a random effects model and the *metafor* R package (*Viechtbauer, 2010*) (available at: https://osf.io/kup8d/). Concordance between the two independent runs (raw Ct values as well as normalized *F. nucleatum* expression) was determined by computing Pearson correlation coefficients (ρ). The original study data was extracted *a priori* from the published figure during preparation of the experimental design. The extracted data was published in the Registered Report (*Repass et al., 2016*) and the published *p*-value was used in the power calculations to determine the sample size for this study.

## Deviations from the registered report

The proposed analysis of age/ethnicity matched samples compared to CRC samples were unable to be performed because of undetectable *F. nucleatum* expression in the matched normal tissue. After an initial attempt the purification and qPCR conditions were optimized in an effort to detect *F. nucleatum* gene expression. This includes re-purification of genomic DNA using Qiagen DNeasy Blood and Tissue Kit and increasing the starting material from 5 ng to 60 or 100 ng of total DNA. To assess if the signal for *F. nucleatum* from the qPCR assay produced a specific product for *F. nucleatum*, we sequenced the amplicons generated as described above. Furthermore, an additional exploratory analysis was performed to test if *F. nucleatum* expression was different between CRC and adjacent normal tissue only in the samples with a detectable *F. nucleatum* signal. Additional materials and instrumentation not listed in the Registered Report, but needed during experimentation are also listed.

# Acknowledgements

The Reproducibility Project: Cancer Biology would like to thank Courtney Soderberg at the Center for Open Science for assistance with statistical analyses and the following companies and labs for generously donating reagents to the Reproducibility Project: Cancer Biology; American Type and Tissue Collection (ATCC), Applied Biological Materials, BioLegend, Charles River Laboratories, Corning Incorporated, DDC Medical, EMD Millipore, Harlan Laboratories, LI-COR Biosciences, Mirus Bio, Novus Biologicals, Sigma-Aldrich, and System Biosciences (SBI).

# Additional information

### Group author details

**Reproducibility Project: Cancer Biology**
Elizabeth Iorns: Science Exchange, Palo Alto, United States; Alexandria Denis: Center for Open Science, Charlottesville, United States; Stephen R Williams: Center for Open Science, Charlottesville, United States; Nicole Perfito: Science Exchange, Palo Alto, United States; Timothy M Errington: Center for Open Science, Charlottesville, United States

### Competing interests

John Repass: ARQ Genetics is a Science Exchange associated lab. Reproducibility Project: Cancer Biology: EI, NP: Employed by and hold shares in Science Exchange Inc.The other authors declare that no competing interests exist.

## Funding

| Funder | Author |
| --- | --- |
| Laura and John Arnold Foundation | Reproducibility Project: Cancer Biology |

The funder had no role in study design, data collection and interpretation, or the decision to submit the work for publication.

## Author contributions

John Repass, Acquisition of data, Analysis and interpretation of data, Drafting or revising the article; Reproducibility Project: Cancer Biology, Analysis and interpretation of data, Drafting or revising the article

## Author ORCIDs

Alexandria Denis (iD) http://orcid.org/0000-0002-1210-2309
Timothy M Errington (iD) http://orcid.org/0000-0002-4959-5143

## Ethics

Human subjects: Patient tissue samples were obtained from iSpecimen (Lexington, Massachusetts) as flash-frozen in liquid nitrogen shortly after harvest. Approval was obtained from the iSpecimen institutional review board (protocol # 2011-332) and shared samples and data were de-identified for this study.

## Decision letter and Author response

Decision letter https://doi.org/10.7554/eLife.25801.011
Author response https://doi.org/10.7554/eLife.25801.012

# Additional files

## Supplementary files

• Transparent reporting form
DOI: https://doi.org/10.7554/eLife.25801.008

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
