## [Decision Letter]

Thank you for submitting your article "Replication Study: *Fusobacterium nucleatum* infection is prevalent in human colorectal carcinoma" for consideration by *eLife*. Your article has been reviewed by three peer reviewers, one of whom served as a Guest Reviewing Editor, and the evaluation has been overseen by Sean Morrison as the Senior Editor. The reviewers have opted to remain anonymous.

This paper reflects a pretty straightforward replication of one piece of the Castellarin 2012 paper. Each reviewer has some comments to be addressed in a revised version.

*Reviewer #1:*

The authors tried to replicate a selected experiment from the paper "*Fusobacterium nucleatum* infection is prevalent in human colorectal carcinoma" (Castellarin et al., 2012). However, the authors obtained undetectable *Fusobacterium nucleatum* DNA by qPCR in all 3 different types of samples, despite the fact that they have no problem to measure the control gene PGT. The authors further discussed several approaches taken by others to combat this issue. It is not clear to me whether the authors have taken one of the existing approaches or just used the measurement for *F. nucleatum* without normalizing using PGT. Since it is an important issue, it would help reviewers to evaluate the conclusion if the authors present in detail how they solved this problem and the rationale of their solutions.

Subsection “Quantitative PCR of *F. nucleatum* DNA abundance from colorectal carcinoma, adjacent normal tissue, and matched normal human colon tissue”. The mean fold change (4.97) is much lower than reported by the original study (378.9). It would be nice to have some discussion on this.

Subsection “Meta-analyses of original and replicated effects” – r = 0.24, 95% CI [-0.08, 0.51] spans 0 indicting that there is no significant correlation. In contrast, the original study showed a significant positive correlation (r= 0.45, 95% CI [0.28, 0.60] (page 157-158). It would be nice to have some discussion on this.

Statistical analysis section: For some reason, I could not find the csv files or analysis scripts on the OSF.

Subsection “Quantitative PCR of *F. nucleatum* DNA abundance from colorectal carcinoma, adjacent normal tissue, and matched normal human colon tissue” – Is it correct that the relative abundance was calculated without using control gene PGT?

Could you please clarify how of the mean normalized expression is calculated, e.g., using control gene PGT?

Please explain how 20-40 is within the normal acceptable range (<30). Perhaps the authors meant that the normal acceptable range is <=40 or 20-30 is within the normal acceptable range (<30).

*Reviewer #2:*

The authors conducted the quantitative PCR of *Fusobacterium nucleatum* DNA abundance from colorectal carcinoma, normal tissue adjacent to carcinoma, and age/ethnicity-matched normal human colon tissue. This is an important topic, given high interest and potentials to expand new study areas. The Materials and methods are well described. The results are clearly demonstrated in the figures, and clearly summarized in the manuscript. There are some useful findings in the present paper. Following points should be considered.

In the literature, some studies used FFPE tissue and others used fresh frozen tissue. That makes a difference in quality of DNA as well as tumor neoplastic cellularity. In general, FFPE tissue can give higher tumor cellularity than fresh frozen tissue if tumor dissection from FFPE tissue is properly performed. When discussing the literature, this point needs to be kept in mind.

At the end of the Results and Discussion section, as the authors appropriately discuss, some unmeasured factors can contribute to difference in results between studies. One such factor may be diet, which can very likely influence the gut microbiota. Recently, a new study has provided evidence for differing influences of fiber-rich diet on incidence of *F. nucleatum*-positive colorectal carcinoma (stronger inhibition than on that of *F. nucleatum*-negative carcinoma) (Mehta et al., 2017). That study by Mehta et al. can be discussed as evidence pre-analytic factors such as diet that may explain the difference in study findings.

The authors may add a table/figure that shows the raw data of Ct in colorectal carcinoma, adjacent normal tissue, and matched normal human colon tissue.

The author may discuss why the point estimate of the replication effect size was not within the 95% confidence interval of the original effect size, while the point estimate of the original effect size was within the 95% CI of the replication effect size. Does this make sense?

*Reviewer #3:*

A few comments for the authors to address:

1) The Materials and methods state that patient phenotypes are available online. However, a description regarding the similarities and differences or unknowns between the original population described by Castellarin and the group assayed in this paper should be provided to the reader.

2) The Introduction of the paper fails to mention one approach that has directly visualized clusters of Fusobacterium associated, at least, with a subset of CRC, i.e., FISH of tumors. This data helps to support the idea that, at least for some patients, *Fusobacterium nucleatum* is tumor-associated. This should be included in the manuscript in this reviewer's opinion.

3) The authors should present the concordance between the 2 qPCR runs conducted. In addition they should comment as to whether the two runs identified the same patients as Fn+ using a Ct cutoff value of 35.

[Editors' note: additional revisions were requested prior to acceptance.]

Thank you for submitting your article "Replication Study: *Fusobacterium nucleatum* infection is prevalent in human colorectal carcinoma" for consideration by *eLife*. Your article has been reviewed by Sean Morrison as the Senior Editor.

The new data has really improved the paper. We're almost there but it's still written in a way that makes it hard to extract the key observations. Below are some suggestions that could be addressed with minor changes to the text that would really improve readability:

1) In the Abstract you wrote "…5% of matched control tissue, 25% of adjacent normal, and 40% of CRC gave a positive signal". My understanding is that the new sequencing data show that many of these "positive signals" didn't actually have detectable *F. nucleatum* because the amplification products were non-specific. If that's true, it's misleading to write that 40% of CRCs gave a positive signal, knowing that many of these samples weren't actually positive for the bacterium. Instead, you should give the numbers of each type of sample that were confirmed to be positive based on sequencing of the amplicon, or at least that generated amplification products within a range of Ct values that were consistent with specific amplification based on the sequencing data.

2) In addition to indicating what fraction of samples of each type were truly positive for *F. nucleatum*, you should indicate in the Abstract whether there was a statistically significant difference in *F. nucleatum* levels among samples in which there was confirmed positivity for *F. nucleatum* in the CRC AND/OR adjacent normal sample.

3) You do a good job of mentioning all of the other studies that compared *F. nucleatum* between CRC and control tissue. However, you usually don't state what fraction of samples from the other studies had detectable *F. nucleatum*. That's the key piece of information that's required to compare your results with theirs. If that is known for some of the other studies (e.g. Gao et al., 2015 and Mima et al., 2016a) that information should be included when you first mention the studies.

4) In subsection “Quantitative PCR of *F. nucleatum* DNA abundance from colorectal carcinoma, adjacent normal tissue, and matched normal human colon tissue”, paragraph five, "this criteria" should be "this criterion".

5) In that same paragraph you should always state the denominator (the total number of samples analyzed) when you state the number that were positive.

6) Rather than going all the way through the paper, with much discussion of positive and negative samples, before presenting the key piece of information why not state up front the number of positive samples, taking into account both the PCR data and the sequencing data. As currently presented, much of the discussion, and the numbers in the Abstract and Results seem misleading when you finally acknowledge that many of the "positive signals" were actually non-specific – with no detectable *F. nucleatum*.

7) In the same section you wrote "In samples with Ct values less than 35…". If it would be accurate to write "In ALL of the samples with Ct values less than 35…" this clarification would be helpful.

8) Figure 1 is unnecessarily unclear about what is presented. Rather than graphing "2-ddCt" why not just show fold change? You should also state explicitly on the Y-axis what is divided by what (CRC/adjacent normal?) rather than leaving it for readers to try to figure out.

---

## [Author Response]

Reviewer #1:

The authors tried to replicate a selected experiment from the paper "Fusobacterium nucleatum infection is prevalent in human colorectal carcinoma" (Castellarin et al., 2012). However, the authors obtained undetectable Fusobacterium nucleatum DNA by qPCR in all 3 different types of samples, despite the fact that they have no problem to measure the control gene PGT. The authors further discussed several approaches taken by others to combat this issue. It is not clear to me whether the authors have taken one of the existing approaches or just used the measurement for F. nucleatum without normalizing using PGT. Since it is an important issue, it would help reviewers to evaluate the conclusion if the authors present in detail how they solved this problem and the rationale of their solutions.

In order to directly compare the replication results to the original study, we performed the normalization (*F. nucleatum* normalized to *PGT*) in a similar manner as the original study despite the concerns raised. We have revised the manuscript (Abstract and relevant section of the Results/Discussion section) to further explain our approach. Further we have reordered the paragraphs to first describe the approach described here before discussing approaches taken by others.

Subsection “Quantitative PCR of F. nucleatum DNA abundance from colorectal carcinoma, adjacent normal tissue, and matched normal human colon tissue”. The mean fold change (4.97) is much lower than reported by the original study (378.9). It would be nice to have some discussion on this.

We have revised the manuscript to include some discussion on these differences.

Subsection “Meta-analyses of original and replicated effects” – r = 0.24, 95% CI [-0.08, 0.51] spans 0 indicting that there is no significant correlation. In contrast, the original study showed a significant positive correlation (r= 0.45, 95% CI [0.28, 0.60] (page 157-158). It would be nice to have some discussion on this.

We have added additional discussion in the meta-analysis section of the revised manuscript.

Statistical analysis section: For some reason, I could not find the csv files or analysis scripts on the OSF.

The files (all data files, analysis/figure scripts, etc) associated with this study are currently private. They will be made public at the time of publication.

Subsection “Quantitative PCR of F. nucleatum DNA abundance from colorectal carcinoma, adjacent normal tissue, and matched normal human colon tissue”. Is it correct that the relative abundance was calculated without using control gene PGT?

The relative abundance was calculated by normalizing the *F. nucleatum* expression to *PGT*, similar to the original study. We revised the manuscript to make this more explicit.

Could you please clarify how of the mean normalized expression is calculated, e.g., using control gene PGT?

We revised the manuscript to include that *F. nucleatum* expression was normalized to *PGT*.

Please explain how 20-40 is within the normal acceptable range (<30). Perhaps the authors meant that the normal acceptable range is <=40 or 20-30 is within the normal acceptable range (<30).

The range reported (~20-40) is for all Ct values observed (*PGT* and *F. nucleatum)*, which are not in the normal acceptable range of less than 30. While the *PGT* Ct values were in the normal range, the *F. nucleatum* was not. We have made this more explicit in the revised manuscript.

Reviewer #2:

The authors conducted the quantitative PCR of Fusobacterium nucleatum DNA abundance from colorectal carcinoma, normal tissue adjacent to carcinoma, and age/ethnicity-matched normal human colon tissue. This is an important topic, given high interest and potentials to expand new study areas. The Materials and methods are well described. The results are clearly demonstrated in the figures, and clearly summarized in the manuscript. There are some useful findings in the present paper. Following points should be considered.In the literature, some studies used FFPE tissue and others used fresh frozen tissue. That makes a difference in quality of DNA as well as tumor neoplastic cellularity. In general, FFPE tissue can give higher tumor cellularity than fresh frozen tissue if tumor dissection from FFPE tissue is properly performed. When discussing the literature, this point needs to be kept in mind.

Thank you for raising this point. We have highlighted this aspect in the revised manuscript.

At the end of the Results and Discussion section as the authors appropriately discuss, some unmeasured factors can contribute to difference in results between studies. One such factor may be diet, which can very likely influence the gut microbiota. Recently, a new study has provided evidence for differing influences of fiber-rich diet on incidence of F. nucleatum-positive colorectal carcinoma (stronger inhibition than on that of F. nucleatum-negative carcinoma) (Mehta et al., 2017). That study by Mehta et al. can be discussed as evidence pre-analytic factors such as diet that may explain the difference in study findings.

Thank you for bringing this to our attention. We have included this in the revised manuscript.

The author may add a table/figure that shows the raw data of Ct in colorectal carcinoma, adjacent normal tissue, and matched normal human colon tissue.

We included distribution plots of the raw Ct counts for *PGT* and *F. nucleatum* for the three tissue types we analyzed. They are displayed in Figure 1—figure supplement 1 (for the first qPCR run) and Figure 1—figure supplement 2 (for the second qPCR run).

The author may discuss why the point estimate of the replication effect size was not within the 95% confidence interval of the original effect size, while the point estimate of the original effect size was within the 95% CI of the replication effect size. Does this make sense?

We have added additional discussion in the meta-analysis section of the revised manuscript.

Reviewer #3:

A few comments for the authors to address:1) The Materials and methods state that patient phenotypes are available online. However, a description regarding the similarities and differences or unknowns between the original population described by Castellarin and the group assayed in this paper should be provided to the reader.

The characteristics of the patient population used in the original study were not reported or made available, thus we are unable to compare them to this replication attempt. We have revised the relevant section in the Materials and methods to describe how this is unknown.

2) The Introduction of the paper fails to mention one approach that has directly visualized clusters of Fusobacterium associated, at least, with a subset of CRC, i.e., FISH of tumors. This data helps to support the idea that, at least for some patients, Fusobacterium nucleatum is tumor-associated. This should be included in the manuscript in this reviewer's opinion.

We have included this in the Introduction of the revised manuscript.

3) The authors should present the concordance between the 2 qPCR runs conducted. In addition they should comment as to whether the two runs identified the same patients as Fn+ using a Ct cutoff value of 35.

Thank you for these suggestions. We have included this information in the revised manuscript. There was high concordance of the raw Ct values between both runs, as well as the gene expression. Additionally, the same patients were identified with a Ct cutoff of 35, with the exception of one patient that was over the cutoff for the first run (Ct = 37.1) and just under it for the second run (Ct = 34.9). Interestingly, all patients that were identified as *F. nucleatum* positive in their adjacent normal tissue were also identified positive in their CRC tissue. We have included this additional analysis in the revised manuscript.

[Editors' note: additional revisions were requested prior to acceptance.]

1) In the Abstract you wrote "…5% of matched control tissue, 25% of adjacent normal, and 40% of CRC gave a positive signal". My understanding is that the new sequencing data show that many of these "positive signals" didn't actually have detectable F. nucleatum because the amplification products were non-specific. If that's true, it's misleading to write that 40% of CRCs gave a positive signal, knowing that many of these samples weren't actually positive for the bacterium. Instead, you should give the numbers of each type of sample that were confirmed to be positive based on sequencing of the amplicon, or at least that generated amplification products within a range of Ct values that were consistent with specific amplification based on the sequencing data.

We agree with this suggestion and have revised the Abstract to reflect only the numbers and results of the confirmed positive samples.

2) In addition to indicating what fraction of samples of each type were truly positive for F. nucleatum, you should indicate in the Abstract whether there was a statistically significant difference in F. nucleatum levels among samples in which there was confirmed positivity for F. nucleatum in the CRC AND/OR adjacent normal sample.

We agree with this suggestion and have revised the Abstract to reflect only the statistical analysis performed on the confirmed positive samples from CRC and adjacent normal.

3) You do a good job of mentioning all of the other studies that compared F. nucleatum between CRC and control tissue. However, you usually don't state what fraction of samples from the other studies had detectable F. nucleatum. That's the key piece of information that's required to compare your results with theirs. If that is known for some of the other studies (e.g. Gao et al., 2015 and Mima et al., 2016a) that information should be included when you first mention the studies.

We included most of this information in the Results/Discussion section; however we agree including this information when the studies are first mentioned in the Introduction is more fitting and have revised the manuscript accordingly.

4) In subsection “Quantitative PCR of F. nucleatum DNA abundance from colorectal carcinoma, adjacent normal tissue, and matched normal human colon tissue”, paragraph five, "this criteria" should be "this criterion".

Thank you for catching this error. We have fixed it in the revised manuscript.

5) In that same paragraph you should always state the denominator (the total number of samples analyzed) when you state the number that were positive.

Thank you for the suggestion. We have included the total number of samples analyzed for each of the numbers given in this section.

6) Rather than going all the way through the paper, with much discussion of positive and negative samples, before presenting the key piece of information why not state up front the number of positive samples, taking into account both the PCR data and the sequencing data. As currently presented, much of the discussion, and the numbers in the Abstract and Results seem misleading when you finally acknowledge that many of the "positive signals" were actually non-specific – with no detectable F. nucleatum.

We agree with this suggestion and have revised the Results/Discussion to start with the observation of the Ct values, then discuss the sequencing and the number of positive samples taking into account both the PCR data and the sequencing data, and ending with the comparison to the original study (only using the PCR data), before going into the meta-analysis section. We have also reordered the figures to include the subset analysis (confirmed *F. nucleatum* positive) as Figure 1 and the complete analysis of all samples (just PCR data) as Figure 2.

7) In the same section you wrote "In samples with Ct values less than 35…". If it would be accurate to write "In ALL of the samples with Ct values less than 35…" this clarification would be helpful.

We agree and have revised the text as suggested in the manuscript.

8) Figure 1 is unnecessarily unclear about what is presented. Rather than graphing "2-ddCt" why not just show fold change? You should also state explicitly on the Y-axis what is divided by what (CRC/adjacent normal?) rather than leaving it for readers to try to figure out.

For each Replication Study, we have presented the results in the same way as the original study to provide a direct comparison. This is the way the original study had presented the results; however we agree it could be made clearer and have revised the y-axis of Figure 1 and Figure 2 to better reflect what is being plotted. It is the fold change in *F. nucleatum* abundance (normalized to *PGT*) in CRC versus adjacent normal tissue.